# Nicotinic Acid Adenine Dinucleotide Phosphate (NAADP) Induces Intracellular Ca^2+^ Release through the Two-Pore Channel TPC1 in Metastatic Colorectal Cancer Cells

**DOI:** 10.3390/cancers11040542

**Published:** 2019-04-15

**Authors:** Pawan Faris, Giorgia Pellavio, Federica Ferulli, Francesca Di Nezza, Mudhir Shekha, Dmitry Lim, Marcello Maestri, Germano Guerra, Luigi Ambrosone, Paolo Pedrazzoli, Umberto Laforenza, Daniela Montagna, Francesco Moccia

**Affiliations:** 1Laboratory of General Physiology, Department of Biology and Biotechnology “L. Spallanzani”, University of Pavia, 27100 Pavia, Italy; pawan.faris@yahoo.com; 2Research Centre, Salahaddin University-Erbil, 44001 Erbil, Kurdistan-Region of Iraq, Iraq; mudhir.shekha@su.edu.krd; 3Human Physiology Unit, via Forlanini 6, Department of Molecular Medicine, University of Pavia, 27100 Pavia, Italy; giorgia.pellavio01@universitadipavia.it (G.P.); lumberto@unipv.it (U.L.); 4Laboratory of Immunology Transplantation, Foundation IRCCS Policlinico San Matteo, 27100 Pavia, Italy; federica.ferulli@gmail.com (F.F.); d.montagna@smatteo.pv.it (D.M.); 5Department of Medicine and Health Sciences “Vincenzo Tiberio”, University of Molise, 86100 Campobasso, Italy; francesca.dinezza@gmail.com (F.D.N.); germano.guerra@unimol.it (G.G.); ambroson@unimol.it (L.A.); 6Department of Pathological Analysis, College of Science, Knowledge University, 074016 Erbil, Kurdistan-Region of Iraq, Iraq; 7Department of Pharmaceutical Sciences, Università del Piemonte Orientale, 28100 Novara, Italy; dmitry.lim@uniupo.it; 8Unit of General Surgery, Foundation IRCCS Policlinico San Matteo, 27100 Pavia, Italy; mmaestri@smatteo.pv.it; 9Department of Sciences Clinic-Surgical, Diagnostic and Pediatric, University of Pavia, 27100 Pavia, Italy; 10Medical Oncology, oundation IRCCS Policlinico San Matteo, 27100 Pavia, Italy; p.pedrazzoli@smatteo.pv.it

**Keywords:** NAADP, TPC1, lysosomal Ca^2+^ signalling, cancer, colorectal carcinoma, proliferation

## Abstract

Nicotinic acid adenine dinucleotide phosphate (NAADP) gates two-pore channels 1 and 2 (TPC1 and TPC2) to elicit endo-lysosomal (EL) Ca^2+^ release. NAADP-induced EL Ca^2+^ signals may be amplified by the endoplasmic reticulum (ER) through the Ca^2+^-induced Ca^2+^ release mechanism (CICR). Herein, we aimed at assessing for the first time the role of EL Ca^2+^ signaling in primary cultures of human metastatic colorectal carcinoma (mCRC) by exploiting Ca^2+^ imaging and molecular biology techniques. The lysosomotropic agent, Gly-Phe β-naphthylamide (GPN), and nigericin, which dissipates the ΔpH which drives Ca^2+^ refilling of acidic organelles, caused massive Ca^2+^ release in the presence of a functional inositol-1,4,5-trisphosphate (InsP_3_)-sensitive ER Ca^2+^ store. Liposomal delivery of NAADP induced a transient Ca^2+^ release that was reduced by GPN and NED-19, a selective TPC antagonist. Pharmacological and genetic manipulations revealed that the Ca^2+^ response to NAADP was triggered by TPC1, the most expressed TPC isoform in mCRC cells, and required ER-embedded InsP_3_ receptors. Finally, NED-19 and genetic silencing of TPC1 reduced fetal calf serum-induced Ca^2+^ signals, proliferation, and extracellular signal-regulated kinase and Akt phoshorylation in mCRC cells. These data demonstrate that NAADP-gated TPC1 could be regarded as a novel target for alternative therapies to treat mCRC.

## 1. Introduction

Spatio-temporal intracellular Ca^2+^ signals regulate a multitude of functions, including proliferation, migration, differentiation, gene expression, bioenergetics and survival [1,2,3,4,5,6]. An increase in intracellular Ca^2+^ concentration ([Ca^2+^]_i_) is generated by the opening of distinct Ca^2+^-permeable channels that can be located either in the plasma membrane, such as voltage-gated Ca^2+^ channels, Transient Receptor Potential (TRP) channels, and store-operated channels (SOCs), or in endogenous organelles, such endoplasmic reticulum (ER), Golgi and the acidic vesicles of the endolysosomal (EL) system (EL) [1,5,6,7,8]. The ER represents the largest intracellular Ca^2+^ reservoir and releases Ca^2+^ in response to appropriate stimuli leading to the synthesis of the second messengers, inositol-1,4,5-trisphosphate (InsP_3_) and cyclic ADP-ribose (cADPR), which respectively target InsP_3_ receptors (InsP_3_Rs) and ryanodine receptors (RyRs) [6,9]. InsP_3_Rs and RyRs are both sensitive to cytosolic Ca^2+^ and can activate each other via a Ca^2+^-induced Ca^2+^ release (CICR) mechanism that is able to amplify local Ca^2+^ signals into a global elevation in [Ca^2+^]_i_ [6,9].

The ER, however, is not the sole endogenous Ca^2+^ store in mammalian cells. Acidic EL vesicles, including endosomes and lysosomes, provide an additional Ca^2+^ reservoir that is recruited by extracellular ligands through the generation of the novel Ca^2+^-releasing second messenger, nicotinic acid adenine dinucleotide phosphate (NAADP) [7,10,11]. Acidic organelles contain a relatively high amount of Ca^2+^, i.e., ≈500 µM [12], which is sequestered in a pH-dependent manner through a yet to be identified mechanism, although a Ca^2+^/H^+^ exchanger (CAX) has been recently identified in non-placental mammalian cells [13,14,15]. NAADP mediates Ca^2+^ release from acidic stores across the phylogenetic tree by activating a novel superfamily of voltage-gated ion channels, i.e., two-pore channels 1–3 (TPC1-3) [16,17,18,19]. However, only TPC1 and TPC2 are present in humans, mice and rats. NAADP does not directly target TPC1 and TPC2, but its action requires the interposition of an NAADP-binding protein, which could also mediate NAADP-dependent Ca^2+^ release from ER-embedded RyRs [20,21]. NAADP-induced mobilization of the acidic Ca^2+^ pool may be then amplified by InsP_3_Rs and RyRs through the mechanism of CICR at the very narrow (<30 nm) membrane contact sites (MCSs) that occur between EL and ER vesicles [12,14,22]. TPC1 is widely distributed across the EL system, while TPC2 mainly resides at late endosomes and lysosomes [16,19,23]. Either NAADP, TPCs or a combination of both control a growing number of processes, including fertilization, insulin secretion, angiogenesis, vasculogenesis, nitric oxide (NO) release, autophagy, and neurite extension [24,25,26,27,28,29,30,31], and have been involved in the pathogenesis of several diseases [10,32].

Recent work showed that TPCs are also involved in tumorigenesis [33,34]. For instance, TPC1 transcripts were approximately three to eight folds higher than TPC2 in the SKBR3 cells, a human breast cancer cell line [18]. Subsequently, it was found that TPC1 and TPC2 transcripts were similarly expressed in a different set of human breast cancer cells, although genetic silencing of either channel did not affect proliferation in the highly aggressive MDA-MB-468 cell line [35]. More recently, TPC1 and TPC2 transcripts were identified in several cancer cell lines established from the liver, bladder and blood [36]. Notably, genetic silencing of TPC1 and TPC2 inhibited bladder cancer cell migration by impairing the formation of leading edges [36]. Moreover, pharmacological blockade of TPCs with the selective antagonist, NED-19, and genetic silencing of TPC2 reduced lung metastasis in a mammary cancer mouse model [36]. Finally, NED-19 impaired tumor growth, vascularization and metastasis in a mouse model of vascular endothelial growth factor (VEGF)-secreting melanoma and reduced VEGF-induced Ca^2+^ signals in B16 melanoma cells [37]. Therefore, TPCs are emerging as a promising molecular target to design alternative anticancer therapies [32,33,34]. 

Herein, we sought to investigate the role of NAADP-induced intracellular Ca^2+^ signaling in primary cultures of metastatic colorectal cancer (mCRC) cells established from liver metastasis of human patients [38,39]. We found that an EL Ca^2+^ store is present and functionally coupled to InsP_3_-dependent ER Ca^2+^ release in mCRC cells. Accordingly, NAADP-induced intracellular Ca^2+^ mobilization was dampened by disrupting both the EL and ER Ca^2+^ pools. NAADP-induced Ca^2+^ release was also blocked by NED-19 and by genetic silencing of TPC1, the major TPC isoform expressed in mCRC cells. Finally, pharmacological and genetic blockade of TPC1 dramatically reduced fetal bovine serum (FBS)-induced Ca^2+^ release and proliferation in mCRC cells, thereby hinting at TPC1 as a novel therapeutic target in mCRC patients.

## 2. Results

### 2.1. A Functional Lysosomal Ca^2+^ Store Is Present in Metastatic Colorectal Cancer Cells

In order to assess the presence of a functional lysosomal Ca^2+^ store, mCRC cells were loaded with the Ca^2+^-sensitive fluorophore, Fura-2/AM, as shown elsewhere [39]. Subsequently, mCRC cells were challenged with the lysosomotropic agent, glycyl-l-phenylalanine-2-naphthylamide (GPN), a substrate of the exopeptidase cathepsin C that results in osmotic lysis and loss of lysosomal membrane integrity [40]. GPN has been widely employed to investigate EL Ca^2+^ signalling in a variety of cell types with the notable exception of cancer cells [14,41,42]. The acute addition of GPN (200 μM) caused a rapid and transient increase in [Ca^2+^]_i_ in the absence of extracellular Ca^2+^ (0Ca^2+^) (Figure 1A), which reflects intracellular Ca^2+^ release from the endogenous lysosomal store [41,43].

To further support this observation, mCRC cells were stained with Lysotracker Red, a fluorescent weak base that accumulates in acidic organelles [43]. As shown in Appendix A, GPN (200 μM, 30 min) caused the loss of Lysotracker Red fluorescence, which is consistent with labelling of lysosomal vesicles. We further probed the effects of the ionophore nigericin, which dissipates the proton gradient sustaining lysosomal Ca^2+^ refilling [15,40]. Similar to GPN, nigericin (50 μM) caused a transient increase in [Ca^2+^]_i_ under 0Ca^2+^ conditions (Figure 1B) and erased Lysotracker Red fluorescence (Appendix A), thereby confirming the EL origin of the intracellular Ca^2+^ signal. As summarized in Figure 1C,D, respectively, there was no significant difference between GPN and nigericin in the percentage of responding cells and in the magnitude of the peak Ca^2+^ response. Taken together, these findings show for the first time that a functional lysosomal Ca^2+^ store is present in mCRC cells and is able to increase the [Ca^2+^]_i_.

### 2.2. InsP_3_-Induced ER Ca^2+^ Release Sustains the Ca^2+^ Response to GPN in Metastatic Colorectal Cancer Cells

As mentioned earlier, lysosomal Ca^2+^ release may be amplified by ER-dependent Ca^2+^ mobilization through the CICR mechanism, which results in the appearance of regenerative Ca^2+^ waves [10,44]. For instance, GPN induces long-lasting intracellular Ca^2+^ oscillations in human fibroblasts that are curtailed by previous depletion of the EL Ca^2+^ store or pharmacological blockade of InsP_3_Rs [41,45]. To challenge this model in mCRC cells, we exploited cyclopiazonic acid (CPA), an established inhibitor of Sarco-Endoplasmic Reticulum Ca^2+^-ATPase (SERCA) activity, which prevents Ca^2+^ sequestration into ER lumen and results in depletion of the ER Ca^2+^ pool. We have recently shown that pretreating mCRC cells with 30 μM CPA for 30 min under 0Ca^2+^ conditions is sufficient to deplete the InsP_3_-sensitive ER Ca^2+^ store in mCRC cells [39]. As shown in Figure 2, this protocol significantly (*p* < 0.05) reduced the percentage of cells responding to GPN (200 μM) (Figure 2A,C) and the peak magnitude of endogenous Ca^2+^ release (Figure 2A,D).

Likewise, CPA (30 μM, 30 min) significantly (*p* < 0.05) reduced the percentage of responding cells and the peak Ca^2+^ response to nigericin (50 μM) (Figure 2B–D). In agreement with these observations, 2-Aminoethoxydiphenyl borate (2-APB) (50 μM, 30 min), which selectively blocks InsP_3_Rs in the absence of extracellular Ca^2+^ [46,47], significantly (*p* < 0.05) reduced the percentage of responding cells (Figure 3B) and the peak Ca^2+^ response to GPN (Figure 3A,C). RyRs could also amplify EL Ca^2+^ release [14]. Caffeine (5 mM), a membrane-permeant agonist of RyRs, did not increase the [Ca^2+^]_i_ in mCRC cells (Appendix A). Moreover, qRT-PCR could not find RyR1-3 transcripts in these cells (Appendix A). These data, therefore, indicate that GPN-induced lysosomal Ca^2+^ mobilization is amplified by InsP_3_-dependent ER Ca^2+^ release through the CICR mechanism, as described elsewhere [41].

### 2.3. NAADP Gates TPC1 to Induce Lysosomal Ca^2+^ Release in Metastatic Colorectal Cancer Cells

In order to assess whether and how NAADP mobilizes lysosomal Ca^2+^, we exploited the NAADP-containing liposomal preparation that we have recently described [48]. As illustrated in Figure 4A, liposomal delivery of NAADP (at 1:20 dilution) caused a rapid Ca^2+^ transient under 0Ca^2+^ conditions, while restoration of extracellular Ca^2+^ did not cause any discernible increase in [Ca^2+^]_i_. NAADP is therefore unlikely to evoke extracellular Ca^2+^ entry in mCRC cells. Control experiments revealed that liposomal formulations devoid of NAADP did not cause any Ca^2+^ signal (Figure 4A). The Ca^2+^ response to NAADP (1:20) was impaired by disruption of acidic stores with GPN (200 μM, 30 min) (Figure 4B), which significantly (*p* < 0.05) reduced both the percentage of responding cells (Figure 4C) and the peak Ca^2+^ response (Figure 4D). 

qRT-PCR showed that TPC1 transcripts were ≈50-folds (50.2 ± 7.4, *n* = 4) higher than TPC2 (Figure 5A), as also found in several cancer cell lines [18,36]. Negative controls were carried out by omitting the reverse transcriptase [39]. 

Accordingly, the Ca^2+^ response to NAADP was significantly (*p* < 0.05) reduced by silencing TPC1 with a selective small interfering RNA (siRNA) (Figure 5B), which significantly (*p* < 0.05) reduced both the percentage of mCRC cells responding to NAADP (1:20) (Figure 5C) and the magnitude of the Ca^2+^ peak (Figure 5D). The efficacy of gene silencing on TPC1 expression has been illustrated in Appendix A. The same blocking effect was achieved by NED-19 (100 μM, 30 min), a selective NAADP antagonist (Figure 5B–D). Collectively, these findings demonstrate that NAADP mobilizes the EL Ca^2+^ store by activating TPC1 in mCRC cells.

### 2.4. InsP_3_-Induced ER Ca^2+^ Release Sustains NAADP-Induced Intracellular Ca^2+^ Signals in Metastatic Colorectal Cancer Cells

As described above, ER-embedded InsP_3_Rs amplify lysosomal Ca^2+^ release via the CICR mechanism. Therefore, we challenged mCRC cells with NAADP-containing liposomes (1:20) upon depletion of the ER Ca^2+^ store with CPA (30 μM, 30 min) under 0Ca^2+^ conditions [39]. As shown in Figure 6A, this treatment significantly (*p* < 0.05) reduced the intracellular Ca^2+^ response to NAADP by diminishing the percentage of activated cells (Figure 6B) and dampening the magnitude of the Ca^2+^ transient (Figure 6C). The same effect was achieved by blocking InsP_3_Rs with 2-APB (50 μM, 30 min), as depicted in Figure 6A and summarized in Figure 6B,C. Overall, these findings endorse the view that InsP_3_-dependent ER Ca^2+^ release sustains the lysosomal Ca^2+^ burst induced by NAADP.

### 2.5. TPC1 Triggers FBS-Induced Intracellular Ca^2+^ Release and Stimulates Proliferation in Metastatic Colorectal Cancer Cells

We have previously shown that 20% FBS causes InsP_3_-dependent ER Ca^2+^ release that in turn results in depletion of ER Ca^2+^ levels and activates SOCs in mCRC cells [39]. As InsP_3_-induced Ca^2+^ release from ER stores may be induced by lysosomal Ca^2+^ mobilization (see Figure 2), we analyzed the Ca^2+^ response to 20% FBS after disruption of acidic stores with GPN (200 μM, 30 min) and nigericin (50 μM, 30 min). These treatments impaired 20% FBS-induced endogenous Ca^2+^ release under 0Ca^2+^ conditions (Figure 7A) by significantly reducing the percentage of responding cells (Figure 7B) and magnitude of the intracellular Ca^2+^ peak (Figure 7C). 

Likewise, the Ca^2+^ response to 20% FBS was dramatically reduced in mCRC cells deficient of TPC1 (Figure 8A–C) or pretreated with NED-19 (100 μM, 30 min) to block NAADP-induced Ca^2+^ mobilization (Figure 8A–C). The increase in [Ca^2+^]_i_ induced by FBS has been shown to drive proliferation in several cancer cell lines [49,50,51]. However, the pharmacological and genetic blockade of SOCs did not affect the proliferation response to 20% FBS in mCRC cells [39]. Conversely, mCRC cell proliferation was significantly (*p* < 0.05) reduced in mCRC cells lacking TPC1 or pretreated with NED-19 (100 μM, 30 min) (Figure 8D). 

In order to assess the Ca^2+^-dependent decoders whereby TPC1 mediates mCRC proliferation, we focused on extracellular-signal related kinases (ERK) and protein kinase B (Akt), which were previously been shown to support NAADP-induced proliferation [25]. The Ca^2+^-dependent phosphorylation of ERK and Akt was investigated as recently shown [3] under control conditions and upon pharmacological blockade of TPC1 with NED-19 (100 μM, 30 min) or genetic deletion of TPC1 with the selective siRNA. As illustrated in Figure 9 and Figure 10, respectively, both treatments significantly (*p* < 0.05) reduced the extent of ERK and Akt phosphorylation, which is consistent with previous reports by Favia and colleagues [25]. Therefore, it is possible to conclude that NAADP promotes mCRC cell growth by stimulating TPC1 to mobilize their lysosomal Ca^2+^ content, thereby recruiting the ERK and phosphoinositide 3-kinase (PI3K)/Akt phosphorylation cascades.

## 3. Discussion

Herein, we demonstrated for the first time that a functional lysosomal Ca^2+^ store is present in mCRC cells and functionally coupled to ER-dependent Ca^2+^ release through InsP_3_Rs. We further showed that the novel second messenger, NAADP, stimulates TPC1 to mobilize the EL Ca^2+^ store, thereby engaging InsP_3_Rs to generate a transient increase in [Ca^2+^]_i_. We finally illustrated that NAADP-dependent TPC1-mediated Ca^2+^ signals underlie FBS-induced endogenous Ca^2+^ release and proliferation in mCRC cells. These data endorse the view that TPCs may provide alternative targets to design novel anticancer therapies and that the therapeutic outcome of their inhibition should be investigated in vivo. 

Recent studies uncovered the role played by TPC1 and TPC2 during tumorigenesis [33,34]. These investigations depicted a general pattern according to which TPC1 is more abundant than TPC2 in the majority of cancer cell lines examined [18,36]. Although the involvement of TPCs in cancer growth, vascularization and metastasis was unequivocally demonstrated [35,36,37], several issues remained to be addressed in cancer cells. First, are acidic EL vesicles able to trigger a global increase in [Ca^2+^]_i_? Second, is the lysosomal Ca^2+^ store functionally coupled to the main ER Ca^2+^ store, as observed in normal cell types? Third, is NAADP per se able to mobilize lysosomal Ca^2+^ through TPCs? Herein, we first demonstrated that a robust endogenous Ca^2+^ release could be evoked by challenging mCRC cells with two structurally and mechanistically unrelated agents which target the lysosomal Ca^2+^ content. Accordingly, both the lysosomotropic compound GPN and the ionophore nigericin induced a transient increase in [Ca^2+^]_i_ under 0Ca^2+^ conditions that was associated to the loss of Lysotracker Red fluorescence. These findings strongly suggested that an acidic Ca^2+^ store was present in mCRC cells and could be depleted to elevate intracellular Ca^2+^ levels. Acidic vesicles may establish quasi-synaptic MCSs with the ER, which confers them with the unique role of triggering physiologically relevant Ca^2+^ signals by inducing ER-dependent Ca^2+^ release via CICR [22,52,53]. In agreement with previous findings in human fibroblasts [41], the Ca^2+^ response to GPN was dramatically halted by depletion of the ER Ca^2+^ store with CPA and pharmacological blockade of InsP_3_Rs with 2-APB. Likewise, CPA dramatically affected nigericin-induced intracellular Ca^2+^ mobilization. These observations confirm that ER-embedded InsP_3_Rs are required to globalize the increase in [Ca^2+^]_i_ evoked by lysosomal Ca^2+^ release also in mCRC cells. Notably, a recent mathematical model confirmed that local lysosomal Ca^2+^ leak may be amplified into a regenerative Ca^2+^ wave by juxtaposed ER-resident InsP_3_Rs [45]. These data collectively demonstrate that a Ca^2+^-dependent functional cross-talk between acidic and ER Ca^2+^ stores also exist in cancer cells, such as mCRC cells. Future work will have to assess the occurrence of MCSs in mCRC cells at structural level.

Subsequently, we used a liposomal preparation to load mCRC cells with the membrane-impermeant NAADP and examine the following increase in [Ca^2+^]_i_. NAADP induced a transient Ca^2+^ signal that was associated to endogenous Ca^2+^ release, but not to extracellular Ca^2+^ entry. To the best of our knowledge, this was the first demonstration that NAADP per se was able to increase the [Ca^2+^]_i_ in a cancer cell line other than cervical cancer HeLa cells [14]. The Ca^2+^ response to NAADP was impaired by disrupting acidic stores with GPN, by genetic silencing of TPC1, and by NED-19, a selective antagonist that prevents NAADP from binding to TPCs [54]. Moreover, our results confirmed previous results obtained in cancer cells by showing that TPC1 was ≈50-fold more abundant than TPC2 transcript. These data, therefore, hint at TPC1 as the molecular target recruited by NAADP to mobilize the acidic Ca^2+^ in mCRC cells and provide the first evidence that NAADP directly triggers intracellular Ca^2+^ signals in cancer cells. As expected, the Ca^2+^ response to NAADP was impaired by depleting the ER Ca^2+^ pool with CPA or by inhibiting InsP_3_Rs with 2-APB. This observation is consistent with the functional coupling between the acidic and ER Ca^2+^ stores described above and demonstrates that InsP_3_Rs sustain NAADP-evoked Ca^2+^ signals also in mCRC cells. Likewise, NAADP-induced Ca^2+^ signals are severely dampened by blocking InsP_3_Rs and/or RyRs in a variety of normal cells, including primary human cytotoxic T lymphocytes [55], rat pulmonary arterial myocytes [56], HeLa cells [14], and mouse pancreatic acinar cells [44]. Our findings, therefore, extend to cancer cells the requirement of ER-dependent Ca^2+^ release for NAADP to trigger a global increase in [Ca^2+^]_i_.

Intracellular Ca^2+^ signals support many cancer hallmarks, including aberrant proliferation, enhanced migration and metastasis, resistance to apoptosis and angiogenesis [5,57,58,59]. Although TPC1 is generally more abundant in cancer cell lines, as mentioned above, a clear role in tumorigenesis has been uncovered for TPC2 [18,35,36,37]. Recent investigations demonstrated that NAADP may deliver Ca^2+^ signals which stimulate cancer cells to grow by recruiting a number of Ca^2+^-dependent decoders, including ERK, Akt and c-Jun N-terminal kinase (JNK) [25]. It has long been known that FBS stimulates cancer cell proliferation and migration by inducing InsP_3_-dependent Ca^2+^ release followed by store-operated Ca^2+^ entry (SOCE) [49,50,51]. However, genetic and pharmacological impairment of Stim1, Stim2, Orai1 and Orai3, which mediate SOCE in mCRC cells, do not affect their proliferation rate [39]. This finding strongly suggests that SOCE is not the Ca^2+^-permeable pathway that underlies the proliferation effect of FBS in mCRC cells. In the present investigation, we provided the evidence that NAADP-induced TPC1 activation triggers 20% FBS-induced intracellular Ca^2+^ release in mCRC cells. This conclusion is supported by the evidence that the intracellular Ca^2+^ response to 20% FBS is impaired (1) by disrupting acidic Ca^2+^ stores with GPN/nigericin and (2) by genetic (with a specific siRNA) and pharmacological (with NED-19) blockade of TPC1-mediated Ca^2+^ release. We, therefore, propose that NAADP triggers the Ca^2+^ response to FBS, which is then amplified by InsP_3_-dependent ER Ca^2+^ release and sustained over time by SOCE activation. Unlike SOCE, however, TPC1 supports mCRC growth as the proliferation rate of mCRC cells is dramatically reduced upon genetic silencing of TPC1 or in the presence of NED-19 to prevent lysosomal Ca^2+^ release. We further showed that pharmacological and genetic blockade of TPC1 remarkably reduced ERK and Akt phosphorylation, which have long been known to support Ca^2+^-dependent proliferation in both normal and neoplastic cells [3,60,61]. The 5-year survival of CRC patients showing disease recurrence or being metastatic at diagnosis falls from 80–90% to 10–20% despite development of novel chemotherapeutics or the introduction of targeted therapies against VEGF and EGF signaling [62]. Therefore, TPC1 may be regarded as a novel promising target to develop alternative treatments for individual suffering from mCRC.

## 4. Materials and Methods

### 4.1. Expansion of Tumor Cells

After signing an informed consent, patients (>18 years) affected by mCRC who had undergone surgery intervention to remove primary tumor and/or liver metastases, were enrolled. All procedures were performed according to the guidelines prescribed for the treatment of CRC neoplasia, and no patient was subjected to unnecessary invasive procedures. The present study was approved by the Foundation IRCCS Policlinico San Matteo (Ethical code 20110000996, 17/01/2011). Tumor specimens were processed as previously described with the GentleMACS Dissociator (Miltenyi Biotec, Bergisch Gladbach, Germany) after being treated, with Tumor dissociation Kit (Miltenyi Biotec, Bergisch Gladbach, Germany), according to the manufacturers’ instructions [38,39]. Tumor cells were filtered to remove clusters, checked for viability with trypan blue die exclusion and resuspended at a concentration of 0.5–1 × 10^6^ cells/mL of CellGro SCGM (Cell Genix, Freiburg, Germany), supplemented with 20% FBS, 2 mM L-glutamine, (complete medium) (Life Technologies Inc, Carlsbad, CA, USA) and cultured in 25 cm^2^ tissue flasks (Corning, Stone Staffordshire, England) at 37 °C and 5% CO_2_. Viable tumor cells attached to the flask within 12–24 h. Cultures at 75% to 100% confluence were selected for subculture by trypsinization with 0.25% trypsin and 0.02% EDTA (Life Technologies Inc) in a calcium/magnesium-free balanced solution. The culture medium was changed twice a week and cellular homogeneity evaluated microscopically every 24–48 h. Cells were cryopreserved in 90% FBS and 10% dimethyl sulfoxide and stored in liquid nitrogen for further experiments. To confirm the neoplastic origin of cultured cells obtained after 3–5 passages underwent to morphological and immunocytochemical analysis [38,63]. To confirm the neoplastic origin of cultured cells, at least 3 cytospins were performed using 1 × 10^5^ cultured cells/cytospin obtained after 4–6 passages, for morphologic and immunocytochemical analysis. Cells were fixed in alcohol 95° and 1 slide was stained with hematoxylin-eosin to identify malignant cells on the basis of cytomorphology. To distinguish the tumor from hyperplastic mesothelial cells, the other slides were tested with monoclonal antibodies against cytokeratin CAM 5.2 (Dako, Glostruo, Denmark) using indirect immunoenzymatic staining according to the manufacturers’ instructions. Appendix A reports two representative images with regard to immunohistochemistry performed to evaluate the neoplastic origin of mCRC cells.

For proliferation assays, tumor cells were thawed and plated at the concentration of 10–20 × 10^5^/mL and evaluated after 3–4 days when reached the optimal confluency, as described in [39]. Results are expressed as mean ± SE under each condition and the cells were obtained from all four patients. Differences were assessed by the Student *t*-test for unpaired values as related to controls. All statistical tests were carried out with GraphPad Prism 4 (San Diego, CA, USA).

### 4.2. Solutions for Intracellular Ca^2+^ Recordings

Physiological salt solution (PSS) had the following composition (in mM): 150 NaCl, 6 KCl, 1.5 CaCl_2_, 1 MgCl_2_, 10 Glucose, 10 Hepes. In Ca^2+^-free solution (0Ca^2+^), Ca^2+^ was substituted with 2 mM NaCl, and 0.5 mM EGTA was added. Solutions were titrated to pH 7.4 with NaOH. The osmolality of PSS as measured with an osmometer (Wescor 5500, Logan, UT, USA) was 338 mmol/kg.

### 4.3. Preparation of NAADP-Containing Liposomes

NAADP-containing liposomes were prepared from lecithin by a thin film hydration method, as recently shown in [48]. A thin film was formed by dissolving the lecithin in chloroform/methanol solution (2:1, v/v) in a round bottom flask and following removal of the solvent under vacuum condition at room temperature, which ensured complete removal of the solvents. The film was then hydrated with PBS buffer (10 mM, pH 7.4) to make a 20 mL of lipid coarse dispersion. Liposomes were prepared by adding cholesterol in a 89:20 lecithin:cholesterol molar ratio, codissolved in chloroform and then dried. The dried film from a flask was suspended in 4 mL of rehydration solution. The resulting liposomal dispersion was sonicated [64] for 3 min (Ultrasound Homogenizer-Biologics) and extruded 21 times with 100 nm filter. Finally, the mixture was dialyzed in PBS bulk for 24 h with 3 bulk-changes. Properties of liposomes were modulated by varying the rehydration solution composition. Liposomes were prepared from PBS solution containing only 70 g of NAADP. Moreover, liposomes free to NAADP were prepared directly from the PBS solution, this type of liposome referred as FL was used as reference. NAADP was diluted at 1:20, as shown elsewhere [48].

### 4.4. [Ca^2+^]_i_ Measurements

mCRC cells were loaded with 4 µM fura-2 acetoxymethyl ester (Fura-2/AM; 1 mM stock in dimethyl sulfoxide) in PSS for 30 min at 37 °C and 5% CO_2_, as shown in [39]. After washing in PSS, the coverslip was fixed to the bottom of a Petri dish and the cells observed by an upright epifluorescence Axiolab microscope (Carl Zeiss, Oberkochen, Germany), usually equipped with a Zeiss ×40 Achroplan objective (water-immersion, 2.0 mm working distance, 0.9 numerical aperture). The cells were excited alternately at 340 and 380 nm, and the emitted light was detected at 510 nm. A first neutral density filter (1 or 0.3 optical density) reduced the overall intensity of the excitation light and a second neutral density filter (optical density = 0.3) was coupled to the 380 nm filter to approach the intensity of the 340 nm light. A round diaphragm was used to increase the contrast. The excitation filters were mounted on a filter wheel (Lambda 10, Sutter Instrument, Novato, CA, USA). Custom software, working in the LINUX environment, was used to drive the camera (Extended-ISIS Camera, Photonic Science, Millham, UK) and the filter wheel, and to measure and plot on-line the fluorescence from 10 up to 40 rectangular “regions of interest” (ROI). Each ROI was identified by a number. Since cell borders were not clearly identifiable, a ROI may not include the whole cell or may include part of an adjacent cell. Adjacent ROIs never superimposed. [Ca^2+^]_i_ was monitored by measuring, for each ROI, the ratio of the mean fluorescence emitted at 510 nm when exciting alternatively at 340 and 380 nm (shortly termed “ratio”). An increase in [Ca^2+^]_i_ causes an increase in the ratio [39]. Ratio measurements were performed and plotted on-line every 3 s. The experiments were performed at room temperature (22 °C).

### 4.5. RNA Isolation and Real Time RT-PCR (qRT-PCR)

Total RNA was extracted from mCRC cells using the QIAzol Lysis Reagent (QIAGEN, Milan, Italy). Single cDNA was synthesized from RNA (1 μg) using random hexamers and M-MLV Reverse Transcriptase (Promega, Milan, Italy). Reverse transcription was always performed in the presence or absence (negative control) of the reverse transcriptase enzyme. qRT-PCR was performed in triplicate using 1 μg cDNA and specific primers (intron-spanning primers) for *TPCN1* and *TPCN2*, and for *RyR1*, *RyR2* and *RyR3* (Appendix A). Briefly, GoTaq qPCR Mastermix (Promega, Milan, Italy) was used according to the manufacturer instruction and qRT-PCR performed using Rotor Gene 6000 (Corbett, Concorde, NSW, Australia). The conditions were as follows: initial denaturation at 95 °C for 5 min; 40 cycles of denaturation at 95 °C for 30 s; annealing at 58 °C for 30 s, and elongation at 72 °C for 40 s. The qRT-PCR reactions were normalized using β-actin (BAC) as housekeeping genes. The triplicate threshold cycles (Ct) values for each sample were averaged resulting in mean Ct values for both the gene of interest and the housekeeping genes. The gene Ct values were then normalized to the housekeeping gene by taking the difference: ΔCt = Ct_[gene]_ − Ct_[housekeeping]_, with high ΔCt values reflecting low mRNA expression levels. Melting curves were generated to detect the melting temperatures of specific products immediately after the PCR run. However, PCR products were also separated with agarose gel electrophoresis, stained with ethidium bromide. The molecular weight of the PCR products was compared to the DNA molecular weight marker VIII (Roche Molecular Biochemicals, Monza, Italy).

### 4.6. Membrane Preparation and Immunoblotting

Cells were diluted in PBS and centrifuged 1000× *g* for 10 min. The cell pellets were resuspended with RIPA buffer (150 mM NaCl, 1.0% Triton X-100, 0.5% sodium deoxycholate, 0.1% SDS, 50 mM Tris-HCl, pH 8.0) containing 0.1 mg/mL PMSF, Protease Inhibitor Cocktails (P8340, Sigma-Aldrich Inc., Milan, Italy) and Phosphatase Inhibitor (1 mM Sodium Orthovanadate). The homogenates were solubilized in Laemmli buffer [65] and 30 µg proteins were separated on precast gel electrophoresis (4–20% Mini-PROTEAN TGX Stain-Free Gels, Bio-Rad, Milan, Italy) and transferred to the Hybond-P PVDF Membrane (GE Healthcare, Milan, Italy) by electroelution. After 1 h blocking with Tris buffered saline (TBS) containing 3% BSA and 0.1% Tween (blocking solution) the membranes were incubated overnight at 4 °C with the following antibodies: affinity purified rabbit anti-TPC1 (SAB2104213; Sigma-Aldrich In.), phospho-Akt (p-Akt; Cell Signaling Technology, 4060, Pero (Mi), Italy) or phospho-Erk (p-Erk; Cell Signaling Technology, 4377), diluted 1:1000 in 3% BSA in T-TBS. The membranes were washed and incubated for 1 h with peroxidase-conjugated goat anti-rabbit IgG (Chemicon, AP132P, Merck Millipore, Milan, Italy) or peroxidase-conjugated rabbit anti-mouse IgG (Dakocytomation, P0260, Agilent, Cernusco sul Naviglio (Mi), Italy), diluted 1:100,000 in blocking solution. The bands were detected with ECL™ Select western blotting detection system (GE Healthcare Europe GmbH, Italy). Prestained molecular weight markers (ab116028, Abcam-Prodotti Gianni, Milan, Italy) were used to estimate the molecular weight of the bands. Blots were stripped as shown in and re-probed with RabMAb anti β-2-microglobulin antibody ([EP2978Y] ab75853; ABCAM) as housekeeping. The antibody was diluted 1:10,000 in blocking solution.

### 4.7. Protein Content

Protein contents of all the samples were determined by the Bradford’s [66] method using bovine serum albumin (BSA) as standard. 

### 4.8. Gene Silencing

siRNA targeting TPC1 was purchased by Sigma-Aldrich Inc. MISSION esiRNA (human TPCA1, EHU016301). Scrambled siRNA was used as negative control. Briefly, once the monolayer cells had reached 50% confluency, the medium was removed and the cells were added with Opti-MEM I reduced serum medium without antibiotics (Life Technologies, Milan, Italy). siRNAs (100 nM final concentration) were diluted in Opti-MEM I reduced serum medium and mixed with Lipofectamine™ RNAiMAX transfection reagent (Life Technologies, Milan, Italy) pre-diluted in Opti-MEM), according to the manufacturer’s instructions. After 20 min incubation at room temperature, the mixes were added to the cells and incubated at 37 °C for 5 h. Transfection mixes were then completely removed and fresh culture media was added. The effectiveness of silencing was determined by immunoblotting (see Appendix A) and silenced cells were used 48 h after transfection.

### 4.9. Lysotracker Red Staining

Lysosomal compartment was visualized in vital cells by incubating the cultures in the presence of LysoTraker Red, as described in [67]. Briefly, cells were incubated with Lysotracker Red cells were incubated with (50 nM) for 20 min at 37 °C. To visualize lysosomes, a BX51 Olympus microscope (Segrate (Mi), Italy) equipped with a 100 W mercury lamp was used with the following configuration: 540 nm (excitation filter), 580 nm (dichroic filter) and 590 nm (barrier filter).

### 4.10. Statistics

All the data have been collected from mCRC cells deriving from at least three coverslips from three independent experiments. The amplitude of intracellular Ca^2+^ release in response to each agonist (NAADP and FBS) or drug (GPN and nigericin) was measured as the difference between the ratio at the peak of intracellular Ca^2+^ mobilization and the mean ratio of 1 min baseline before the peak. Pooled data are given as mean ± SE and statistical significance (*p* < 0.05) was evaluated by the Student’s *t*-test for unpaired observations. Data relative to Ca^2+^ signals are presented as mean ± SE, while the number of cells analyzed is indicated in the corresponding bar histograms.

### 4.11. Chemicals

Fura-2/AM and Lysotracker Red were obtained from Molecular Probes (Molecular Probes Europe BV, Leiden, The Netherlands). All the chemicals were of analytical grade and obtained from Sigma Chemical Co. (St. Louis, MO, USA).

## 5. Conclusions

In conclusion, this study demonstrated for the first time that a functional lysosomal Ca^2+^ store is present and functionally coupled to ER-dependent Ca^2+^ release through InsP_3_Rs in mCRC cells. NAADP stimulates TPC1 to mobilize lysosomal Ca^2+^ and mediate 20% FBS-induced mCRC proliferation. TPC1 stimulates proliferation by recruiting the Ca^2+^-dependent ERK and PI3K/Akt phoshorylation cascades. These data extend to cancer cells previous findings on normal cells and lend further support to the notion that NAADP-induced TPC activation could be targeted to eradicate secondary metastasis in cancer patients.

## Figures and Tables

**Figure 1 cancers-11-00542-f001:**
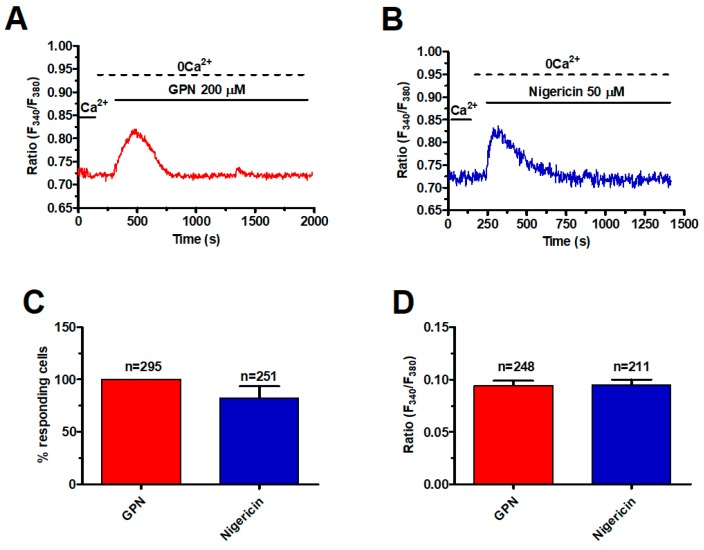
A functional lysosomal Ca^2+^ store is present in metastatic colorectal cancer (mCRC) cells. (**A**) the lysosomotropic compound, glycyl-l-phenylalanine-2-naphthylamide (GPN) (200 µM), caused a robust increase in [Ca^2+^]_i_ in the absence of external Ca^2+^ (0Ca^2+^) in mCRC cells loaded with the Ca^2+^-sensitive fluorophore, Fura 2/AM (4 µM, 30 min). (**B**) nigericin (50 µM), which dissipates the ΔpH that drives Ca^2+^ refilling of acidic organelles, induced a transient elevation in [Ca^2+^]_i_ in mCRC cells. (**C**) mean ± SE of the percentage of responding cells to GPN and nigericin, respectively. (**D**) mean ± SE of the amplitude of the peak Ca^2+^ response to GPN and nigericin, respectively.

**Figure 2 cancers-11-00542-f002:**
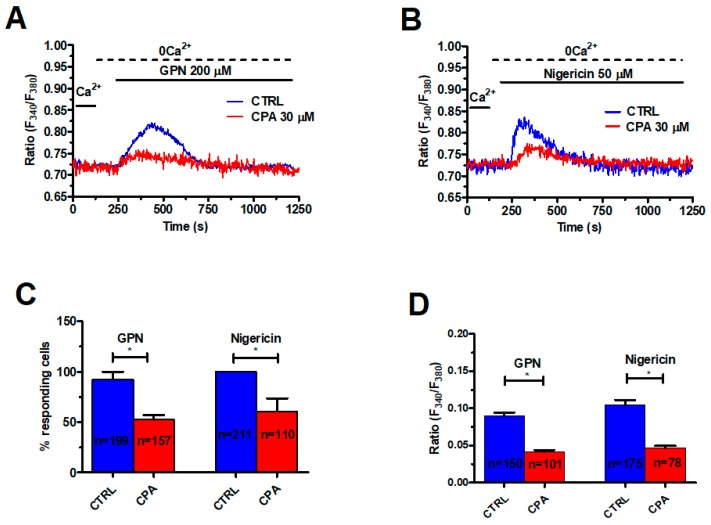
The endoplasmic reticulum (ER) Ca^2+^ store supports the Ca^2+^ response to GPN and nigericin. (**A**) depletion of the ER Ca^2+^ store with cyclopiazonic acid (CPA) (30 μM, 30 min) impaired GPN-induced Ca^2+^ signals in mCRC cells. GPN was administered at 200 µM. (**B**) CPA (30 μM, 30 min) also impaired the transient Ca^2+^ response to nigericin (50 μM). (**C**) mean ± SE of the percentage of responding cells under the designated treatments. The asterisk indicates *p* < 0.05. (**D**) mean ± SE of the amplitude of the Ca^2+^ response to GPN and nigericin under the designated treatments. The asterisk indicates *p* < 0.05.

**Figure 3 cancers-11-00542-f003:**
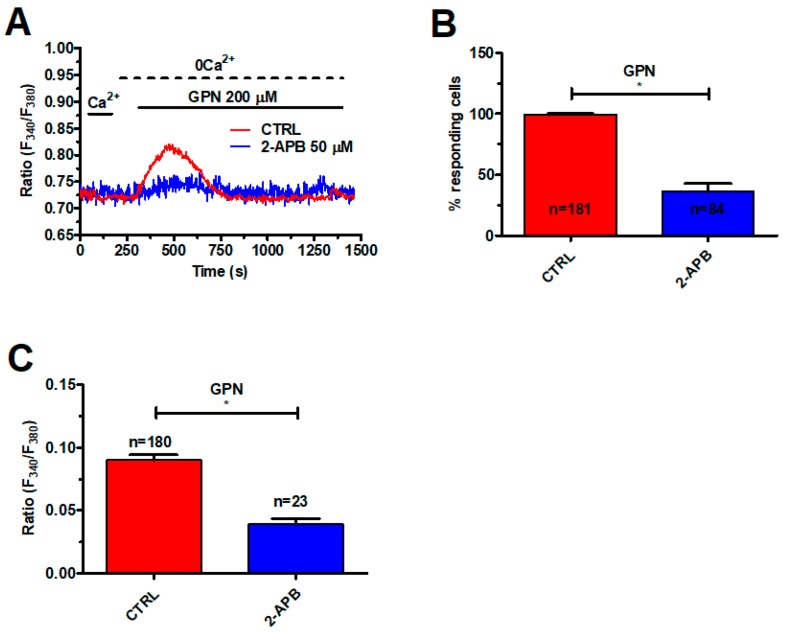
InsP_3_Rs support the Ca^2+^ response to GPN. (**A**) 2-Aminoethoxydiphenyl borate (2-APB) (50 μM, 30 min), a selective InsP_3_R blocker under 0Ca^2+^ conditions, impaired the Ca^2+^ response to GPN (200 µM) in mCRC cells. (**B**) mean ± SE of the percentage of mCRC cells responding to GPN in the absence and in the presence of 2-APB. The asterisk indicates *p* < 0.05. (**C**) mean ± SE of the amplitude of the Ca^2+^ response to GPN in the absence and in the presence of 2-APB. The asterisk indicates *p* < 0.05.

**Figure 4 cancers-11-00542-f004:**
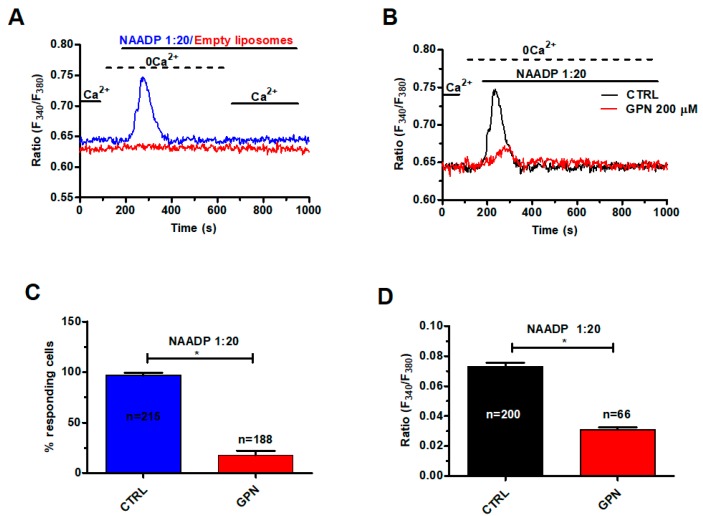
NAADP induces lysosomal Ca^2+^ release in mCRC cells. (**A**) liposomal delivery of NAADP (1:20 dilution) caused a transient elevation in [Ca^2+^]_i_ in the absence of external Ca^2+^ (0Ca^2+^), while restoration of extracellular Ca^2+^ did not cause any additional Ca^2+^ signal. Control liposomes did not induce and detectable increase in [Ca^2+^]_i_. (**B**) disrupting the lysosomal Ca^2+^ store with GPN (200 µM, 30 min) severely affected the Ca^2+^ response to NAADP (1:20 dilution). (**C**) mean ± SE of the percentage of responding cells in the absence and in the presence of GPN. The asterisk indicates *p* < 0.05. (**D**) mean ± SE of the amplitude of the peak Ca^2+^ response to NAADP in the absence and in the presence of GPN. The asterisk indicates *p* < 0.05.

**Figure 5 cancers-11-00542-f005:**
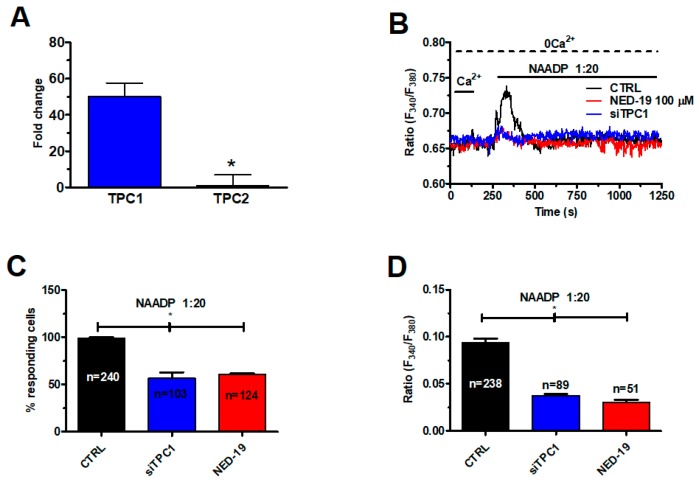
Two-pore channel 1 (TPC1) mediates NAADP-induced lysosomal Ca^2+^ release in mCRC cells. (**A**) qRT-PCR analysis of TPCs revealed that TPC1 transcripts are more abundant as compared to those encoding for TPC2. Data are expressed as Fold change (mean ± SE) of qRT-PCR runs performed in triplicate. The asterisk indicates *p* < 0.05. (**B**) the Ca^2+^ response to liposomal delivery of NAADP (1:20) NAADP was inhibited by NED-19 (100 μM, 30 min), a selective TPC inhibitor, and by the genetic silencing of TPC1 by a specific siRNA. (**C**) mean ± SE of the percentage of responding cells under the designated treatments. The asterisk indicates *p* < 0.05. (**D**) mean ± SE of the amplitude of the peak Ca^2+^ response to NAADP under the designated treatments. The asterisk indicates *p* < 0.05.

**Figure 6 cancers-11-00542-f006:**
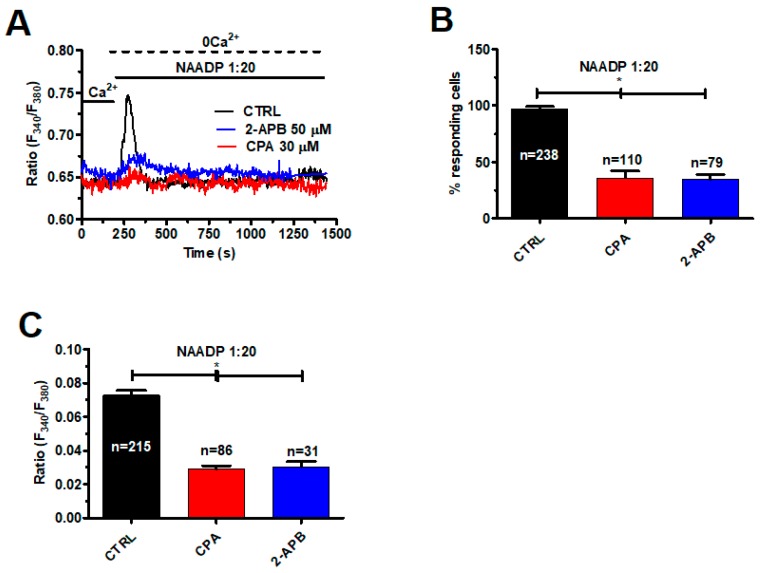
InsP_3_Rs support the Ca^2+^ response to NAADP. (**A**) the intracellular Ca^2+^ response to liposomal delivery of NAADP (1:20) was dramatically reduced by pharmacological depletion of the ER Ca^2+^ pool with CPA (30 μM, 30 min) and by pharmacological blockade of InsP_3_Rs with 2-APB (50 µM, 30 min). (**B**) mean ± SE of the percentage of responding cells under the designated treatments. The asterisk indicates *p* < 0.05. (**C**) mean ± SE of the amplitude of the peak Ca^2+^ response to NAADP under the designated treatments. The asterisk indicates *p* < 0.05.

**Figure 7 cancers-11-00542-f007:**
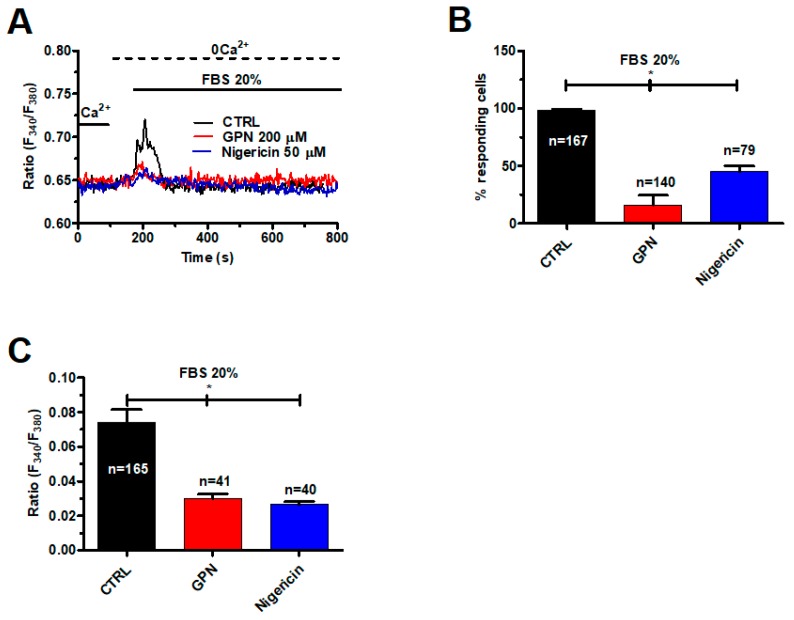
Fetal bovine serum (FBS)-induced intracellular Ca^2+^ release is impaired by disruption of acidic stores in mCRC cells. (**A**) intracellular Ca^2+^ signals induced by FBS 20% were dramatically reduced upon depletion of the lysosomal Ca^2+^ pool with either GPN (200 μM, 30 min) or nigericin (50 μM, 30 min). (**B**) mean ± SE of the percentage of responding cells under the designated treatments. The asterisk indicates *p* < 0.05. (**C**) mean ± SE of the amplitude of the peak Ca^2+^ response to NAADP under the designated treatments. The asterisk indicates *p* < 0.05.

**Figure 8 cancers-11-00542-f008:**
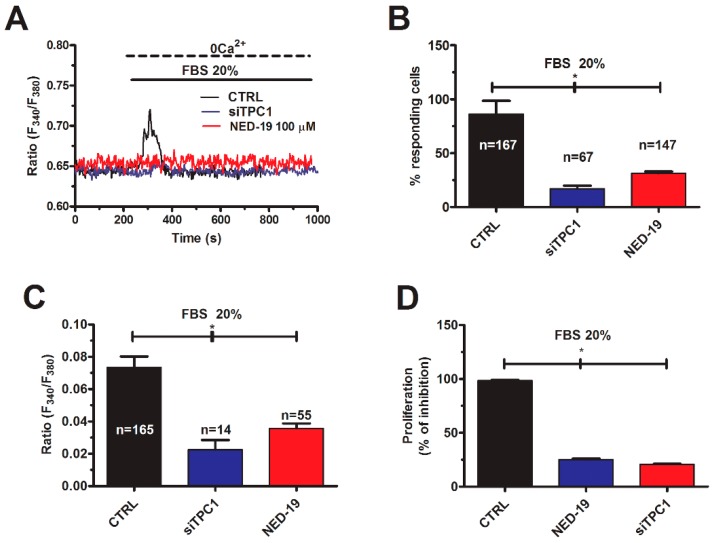
TPC1 mediates FBS-induced lysosomal Ca^2+^ release and proliferation in mCRC cells. (**A**) 20% FBS induced an intracellular Ca^2+^ transient that was significantly reduced by NED-19 (100 μM, 30 min) and by deleting TPC1 with the specific siTPC1. (**B**) mean ± SE of the percentage of responding cells under the designated treatments. The asterisk indicates *p* < 0.05. (**C**) mean ± SE of the amplitude of the peak Ca^2+^ response to NAADP under the designated treatments. The asterisk indicates *p* < 0.05. (**D**) mean ± SE of the percentage of 20% FBS-induced cell proliferation under control conditions and upon pharmacological (NED-19) and genetic (siTPC1) blockade of NAADP-induced Ca^2+^ release. The asterisk indicates *p* < 0.05.

**Figure 9 cancers-11-00542-f009:**
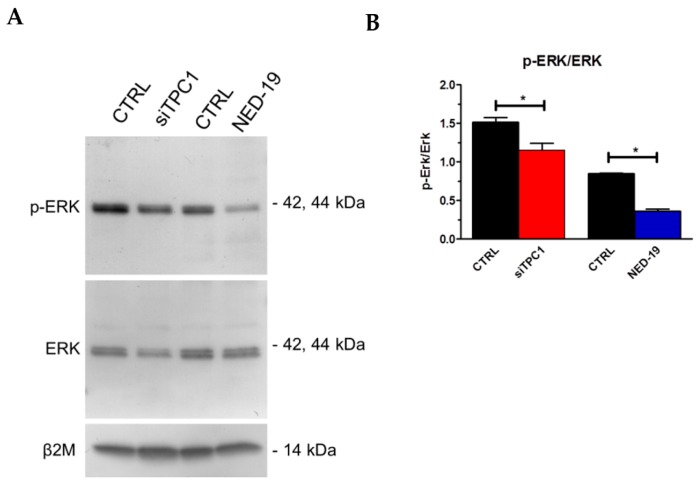
TPC1 stimulates extracellular-signal related kinases (ERK) phosphorylation in mCRC cells. p-ERK and ERK (Aa) bands in mCRC silenced for TPC1 or treated with NED-19 (100 μM, 30 min). Blots representative of four were shown. Lanes were loaded with 30 μg of proteins, probed with affinity purified antibodies and processed as described in Materials and Methods. The same blots were stripped and re-probed with anti-β-2-microglobulin (β2M) antibody, as housekeeping. Major bands of the expected molecular weights were shown (**A**). Bands were acquired, the densitometric analysis of the bands was performed by Total Lab V 1.11 computer program (Amersham Biosciences Europe, Italy) and the results were normalized to non-phosphorylated ERK (**B**). * *p* < 0.05 versus the respective control (Student’s *t* test). The asterisk indicates *p* < 0.05.

**Figure 10 cancers-11-00542-f010:**
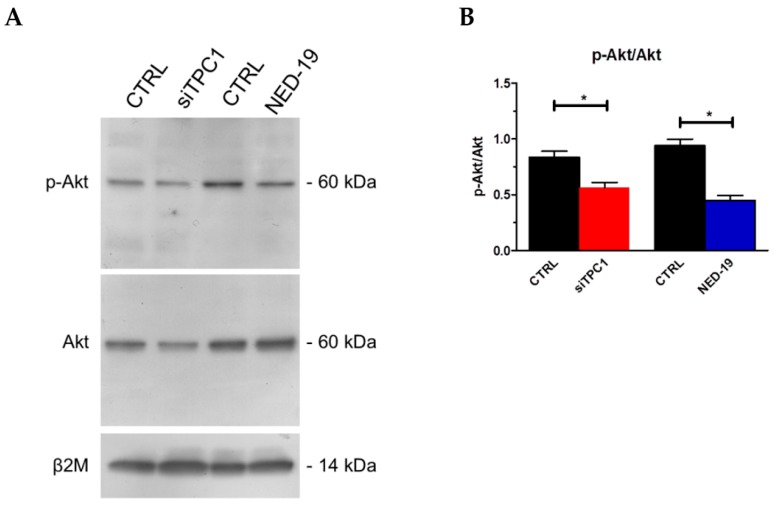
TPC1 stimulates Akt phosphorylation in mCRC cells. p-Akt and Akt (**A**) (Ba) bands in mCRC silenced for TPC1 or treated with NED-19 (100 μM, 30 min). Blots representative of four were shown. Lanes were loaded with 30 μg of proteins, probed with affinity purified antibodies and processed as described in Materials and Methods. The same blots were stripped and re-probed with anti-β2M antibody, as housekeeping. Major bands of the expected molecular weights were shown. Bands were acquired, the densitometric analysis of the bands was performed by Total Lab V 1.11 computer program (Amersham Biosciences Europe, Italy) and the results were normalized to non-phosphorylated Akt (**B**). * *p* < 0.05 versus the respective control (Student’s *t* test). The asterisk indicates *p* < 0.05.

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
