# Peer review of "Nicotinic Acid Adenine Dinucleotide Phosphate (NAADP) Induces Intracellular Ca2+ Release through the Two-Pore Channel TPC1 in Metastatic Colorectal Cancer Cells"

_cancers, 2019, doi:10.3390/cancers11040542_

Round 1

Reviewer 1 Report

No comments

Reviewer 2 Report

The paper is now suitable for publication in its current form

This manuscript is a resubmission of an earlier submission. The following is a list of the peer review reports and author responses from that submission.

Round 1

Reviewer 1 Report

The authors assess for the role of EL Ca2+ signaling in primary cultures of human metastatic colorectal carcinoma (mCRC). The lysosomotropic agent, Gly-Phe β-naphthylamide (GPN), and nigericin have been found to caused massive Ca2+ release in the presence of a full inositol-1,4,5-trisphosphate (InsP3)-sensitive ER Ca2+ store. Moreover, liposomal delivery of NAADP induce a transient Ca2+ release reduced by GPN and NED-19 TPC antagonists. The Ca2+ response to NAADP was triggered by TPC1 and required ER-embedded InsP3 receptors. Finally, NED-19 and genetic silencing of TPC1 reduced fetal calf serum-induced Ca2+ signals, proliferation, and extracellular signal-regulated kinase and Akt phoshorylation in mCRC cells.

The paper is well written, English is fine, however molecular methods are not adequately, reagents e.g., Abs are lacking and selection of patients from primary cell line confusing. mCRC cells derive from one or more mCRC patients? In case or more clones derived from different CRC patients, results relative at these different primary cell line must to be reported. 

Major points      

The major points regard the RNA interference experiments:

1) no data at mRNA levels on the reduction of TPC1 has been reported

2) WB of TPC1 protein levels is very poor and semiquantitative analysis cannot be made.

3) Which anti-TPC1 Ab has been used in western blot? No data on sorces (mouse etc), specificity, (anti-human, -mouse) or other information have been provided in MM or in the figures

4) the experiments in Fig. 5, 6, 7, 8, 9 using a siRNA TPC1 models are lacking of relative scrambled siRNA controls

In addition:

Fig. 6B and 7B,D: which is the mechanism/s of the reductions of responsive cells? Cells are died?

Fig.9 WB of pERK and pAKT is very poor. The levels of total ERK and Akt proteins must to be evaluated. The reduction of ERK and Akt phosphorylation may be depend on a reduction of ERK and Akt protein levels. The ratio pERK/ERK and pAkt/Akt must to be calculated.

Minor points

Suppl Fig. 2 is lacking

Suppl Tab. 1 is lacking

Data relative to histochemical analysis of mCRC cells should be presented in supplemental mats

Discussion is too long and some data are only speculative

So for all considerations the paper is rejected.

Reviewer 2 Report

In this manuscript submitted by Faris et al., authors analyze the role of endo-lysosomal Ca2+ signaling in primary cultures of human metastatic colorectal carcinoma by using fluorescence microscopy (Ca2+ imaging) and molecular biology. Still being a preliminary study, data obtained suggest that two-pore channels 1 (stimulated by the second messenger NAADP) could be a potential target for therapies to treat metastatic colorectal carcinoma. The introduction, results and discussion are well written and referenced.

The experiments in this manuscript appear to be carefully done and they are nicely described. However, the authors might want to consider only a major point:

-      In my opinion the Western blots in Figure 9 must be improved. Both antibodies used for p-Akt (Cell Signaling Technology, 4060) and p-Erk (p-Erk; Cell Signaling Technology, 4377) detection work very well and with 30 ug of total protein loaded, I am sure that the authors will obtain 2 excellent WB. In particular, it is very strange that p44/42 (ERK) Ab does not show you a double band. In addition, you should incubate the membrane with the total (pan) AKT and ERK antibodies to be sure that the protein levels are the same in all the samples. Besides, I do not understand why you strip the membrane. You can easily avoid this step and replace beta-actin with another housekeeping such as GAPDH (37 kDa).

-      I am convinced that WB of Supplementary Figure 3 also can be improved. The quality of these WB is too low, being the background very high and is hard to understand how the authors have quantified these bands.

Minor points:

-       Lines 373: Please replace with “0.5-1x106 cells/ml”

-       Line 376: Please change “hours“ with the symbol “h”

-       Line 383: 10-20 x 105/ml

-       Line 408: DMSO instead of dimethyl sulfoxide

-       Line 447: 150mM should be 150 mM

-       Line 446: (Bradford, 1976). Please add this reference.

-       References: Please check references 31, 32, 48, 49

-       Supplementary material:

     Fig 1: Scale bars ?

     Supplementary Figure 3 should be Supplementary Figure 2